# CRISPR-mediated multiplexed live cell imaging of nonrepetitive genomic loci with one guide RNA per locus

Patricia A. Clow ⓘ [1], Menghan Du[1,2], Nathaniel Jillette[1], Aziz Taghbalout[1], Jacqueline J. Zhu ⓘ [1,3✉] & Albert W. Cheng ⓘ [1,2,3,4,5✉]

Three-dimensional (3D) structures of the genome are dynamic, heterogeneous and functionally important. Live cell imaging has become the leading method for chromatin dynamics tracking. However, existing CRISPR- and TALE-based genomic labeling techniques have been hampered by laborious protocols and are ineffective in labeling non-repetitive sequences. Here, we report a versatile CRISPR/Casilio-based imaging method that allows for a non-repetitive genomic locus to be labeled using one guide RNA. We construct Casilio dual-color probes to visualize the dynamic interactions of DNA elements in single live cells in the presence or absence of the cohesin subunit RAD21. Using a three-color palette, we track the dynamic 3D locations of multiple reference points along a chromatin loop. Casilio imaging reveals intercellular heterogeneity and interallelic asynchrony in chromatin interaction dynamics, underscoring the importance of studying genome structures in 4D.

---

[1] The Jackson Laboratory for Genomic Medicine, Farmington, CT 06032, USA. [2] Department of Genetics and Genome Sciences, University of Connecticut Health Center, Farmington, CT 06030, USA. [3] School of Biological and Health Systems Engineering, Arizona State University, Tempe, AZ 85281, USA. [4] The Jackson Laboratory Cancer Center, Bar Harbor, ME 04609, USA. [5] Institute for Systems Genomics, University of Connecticut Health Center, Farmington, CT 06030, USA. ✉email: jacquelinejufenzhu@gmail.com; albert@cheng.bio

Three-dimensional (3D) organization of the genome has pervasive roles in the regulation of transcription, DNA replication, and DNA repair[1–3]. Sequencing methods[4,5] can uncover genome-wide chromatin interactions in bulk cell populations but are unable to resolve intercellular heterogeneity or temporal dynamics (the "fourth dimension") which instead require live cell imaging techniques[6]. Four-dimensional (4D) live cell imaging of genomic loci can be achieved by inserting into the target loci large tandem arrays of sequences that are bound by well-characterized DNA binding domains fused to fluorescent proteins providing important insights into the dynamics of chromatin interaction and transcription[7,8]. However, these techniques require tedious genome engineering and can be confounded by the side effects of disruptive introduction of exogenous sequences at the target loci. Live-cell imaging of the unmodified genome can be conducted with the use of fluorescent proteins fused to programmable DNA binding proteins such as zinc finger proteins (ZFPs) and transcription activator-like effectors (TALEs), but the complicated cloning protocols requiring weeks of development limit their scalability[9–12]. CRISPR/Cas systems offer versatile DNA binding proteins (e.g., Cas9) that can be programmed by a target-specific guide RNA (gRNA), revolutionizing the field of genome biology[13]. Fluorescent proteins or fluorophores recruited by dCas9 or assembled on the gRNA scaffold offer easily programmable platforms for live cell genomic imaging[14–22] (Supplementary Table 1). However, despite current techniques providing robust labeling of repetitive sequences using a single gRNA, imaging of nonrepetitive sequences requires simultaneous delivery of up to tens of different gRNA per target to achieve sufficient signal, precluding broad adoption.

We previously engineered a modular multitasking CRISPR/Cas-based platform called Casilio by combining dCas9 and engineered Pumilio/FBF (PUF)-tethered effectors[23] (Fig. 1). An effector domain carrying out a specific function (e.g., transcription activation) is fused to a unique PUF RNA binding domain that can bind to one or more copies of a specific 8-mer motif inserted to the 3' end of a gRNA (Fig. 1a). PUF domains contain peptide subunits that can each be programmed to recognize an RNA-base by changing amino acids contacting the RNA, thereby allowing designed PUF domains to bind different 8-mer sequences[24,25]. Casilio capitalizes on the versatility of PUF domains to allow simultaneous and orthogonal delivery of effector functions to distinct genome loci, as well as multimerization of effector molecules, for efficient epigenetic editing[23,26] (Fig. 1b).

In this work, we develop Casilio for imaging nonrepetitive DNA sequences with one gRNA per locus. We apply Casilio to visualize dynamic chromatin interactions in the presence or absence of cohesin subunit RAD21 in live cells. Finally, we expand fluorescent palette of Casilio to allow simultaneous labeling of three loci along a chromatin loop to track its folding dynamics.

## Results

**One-gRNA labeling of a nonrepetitive locus by Casilio**. We reasoned that the immense multimerization capability of Casilio will enable concentration of fluorescent proteins at a target locus to achieve sufficient signal using reduced number of target DNA binding sites while the multiplexing capability will allow simultaneous multicolor labeling of distinct loci in the same cell (Fig. 1c, d). The *MUC4* gene on chromosome 3 was imaged in the first and subsequent CRISPR-based imaging studies by tiling gRNAs over the target region[14,18,22]. In addition to nonrepetitive sequences, the *MUC4* gene harbors repetitive sequences allowing

convenient DNA-FISH co-labeling validation using a single oligonucleotide probe[14]. Therefore, we first tested Casilio for imaging nonrepetitive sequences within the *MUC4* gene. We designed gMUC4 which binds a nonrepetitive sequence within *MUC4* and inserted 15 copies of PUF Binding Site c (15xPBSc) to recruit Clover-PUFc. To allow the assessment of the specificity of Casilio, we co-labeled the *MUC4* region with a Cy5-MUC4 DNA fluorescence in situ hybridization (FISH) probe in U2OS and ARPE-19 cells labelled with Casilio/gMUC4 (Fig. 2a–c). Casilio achieves highly specific labeling of the nonrepetitive locus using only one gRNA with 96% and 100% specificity in U2OS and ARPE-19 cells, respectively. Additional support for specificity is that the formation of fluorescent spots is dependent on the presence of dCas9 (Fig. 2d), and that the number of fluorescent spots is consistent with the ploidy of the diploid ARPE-19 (Fig. 2d) and haploid HAP1 (Fig. 2e) cells. Spot formation efficiency was around ~68% for ARPE-19 (Fig. 2d) and ~54% for HAP1 (Fig. 2e). These together demonstrate that robust and specific labeling of nonrepetitive DNA sequence by Casilio requires only one gRNA.

**Tracking of chromatin interactions with dual-color labeling**. To test whether Casilio can be applied to study the dynamics of chromatin interactions in live cells, we selected a chromatin interaction, the *MASP1-BCL6* loop, with 362 kb genomic distance between the interaction anchors from a published RAD21 ChIA-PET experiment performed in ARPE-19 cells (ENCSR110JOO, Michael Snyder lab). We then designed a pair of gRNAs to bind nonrepetitive sequences next to the anchors (Fig. 3a). We used Clover-PUFc to label the 5' anchor (locus A) and PUF9R-iRFP670 to label the 3' anchor (locus B). Live cell microscopy of ARPE-19 cells transfected with dual-color Casilio probe pairs revealed dynamic chromatin interactions at second timescales and showcased a high degree of intercellular heterogeneity and interallelic asynchrony of chromatin interaction at this loop (Fig. 3b, c; Supplementary Video 1; Supplementary Figs. 1, 2a). Such insights cannot be readily obtained via bulk cell sequencing technologies, supporting the advantages of Casilio live-cell imaging for studying the dynamics of chromatin interactions at single cell and single allele level. To provide future users a method to assess the specificity of labeling in live-cell settings, we developed a "competitor" assay. Here, a "cold" specific competitor gRNA without any PUF Binding Sites (PBS) for fluorescent protein recruitment, but overlaps the target, is co-introduced into cells to interfere with the binding of Casilio imaging complexes and the formation of target spots (Fig. 3d). A non-specific competitor gRNA that does not overlap the target (such as GAL4[27]) is used as a control. To denote a competitor gRNA, we prefix the target with "c" to mean cold or competitor. Specific competitor gRNA cA overlapping with *MASP1-BCL6* locus A greatly reduced the formation of the Clover spots compared to non-specific competitor cGAL4 control (Fig. 3e). As expected, the iRFP670 spots formed at the non-interfered location (*MASP1-BCL6* locus B) served as an internal control that was unaffected by either competitor gRNAs. To delineate the surrounding chromatin structure over an expanded *MASP1-BCL6* region, we tracked an additional location 3' (locus R) 312 kb from locus B (Fig. 3f, g). A–B pair distances (mean 1.36 μm, median 0.97 μm) are significantly lower (Mann–Whitney two-sided $p$-value = $6.01 \times 10^{-29}$) than B–R pair distances (mean 1.81 μm, median 1.40 μm), consistent with the ChIA-PET data which shows more extensive interactions between A–B compared to B–R (Fig. 3f, h).

To test if Casilio can be applied to study the function of cohesin complexes in chromatin interactions, we took advantage of an engineered HCT116 cell line with endogenous RAD21 fused to an

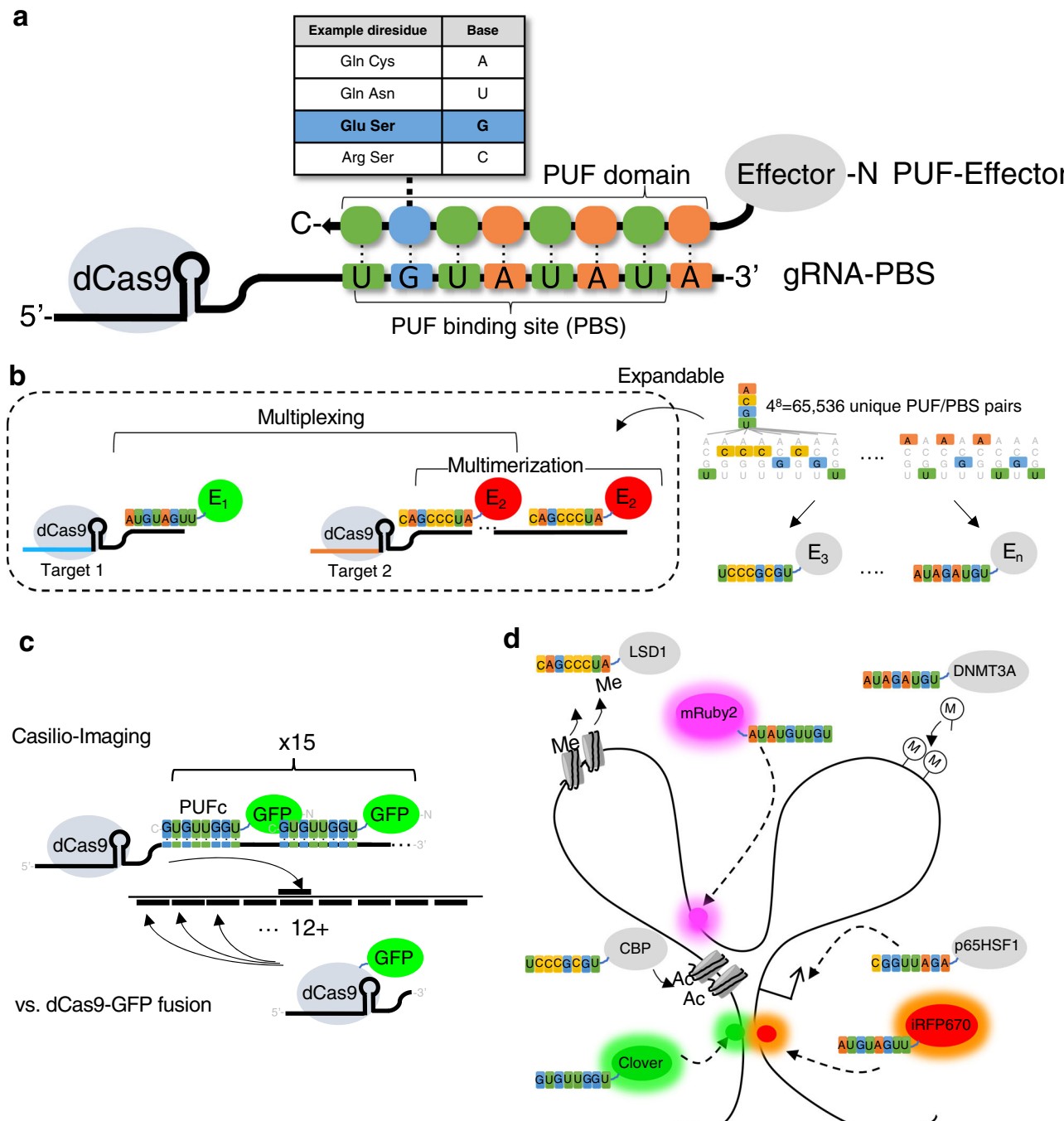

**Fig. 1 Schematic of Casilio. a** In Casilio, gRNA is appended with one or more binding sites for Pumilio/FBF (PUF) RNA binding domain-tethered effectors. Each PUF domain contains 8 peptide subunits each programmable to recognize one of the four RNA bases. A collection of PUF-effectors can be created by fusing different effectors to PUF programmed to recognize different 8-mer PUF binding sequence (PBS). Effectors are recruited to dCas9/gRNA-PBS complex according to the fused PUF domain. **b** Casilio architecture allows multiple molecules of the same or different effectors ($E_1$, $E_2$, ..., etc.) to be recruited to a target (multimerization or complex formation) as well as different set of effectors to be recruited at different targets in the same cell (multiplexing). This system is expandable to potentially over 65,000 possible effectors. **c** Casilio-Imaging capitalizes on the multimerization capability to recruit 15 or more fluorescent proteins to a target as opposed to requiring target site tiling of dCas9-GFP direct fusions using 12 or more individual gRNAs in previous studies. **d** The potential for Casilio to achieve multimodal operations simultaneously, for example, demethylating histone (using LSD1 module), acetylating histone (CBP), methylating DNA (DNMT3A), activating (p65HSF1) different targets, and at the same time monitoring multiple locations with different fluorescent proteins in live cells. dCas9/gRNA-PBS not drawn for simplicity.

auxin-inducible degron (HCT116/RAD21-mAID)[28] that allows for the conditional depletion of RAD21 by auxin treatment. A previous study performed Hi-C assays on this cell line and demonstrated that the depletion of RAD21 resulted in loss of loop domains including a ~500 kb loop domain between the *IER5L* promoter (IER5L-P) and

its super-enhancer (IER5L-SE)[29]. We used Clover-PUFc and PUF9R-iRFP670 to label the *IER5L* promoter and super-enhancer, respectively (Fig. 4a; Supplementary Fig. 3), and performed live-cell microscopy on untreated (-Auxin) (Fig. 4b; Supplementary Video 2; Supplementary Figs. 2b, 4) and RAD21-depleted (+Auxin) cells

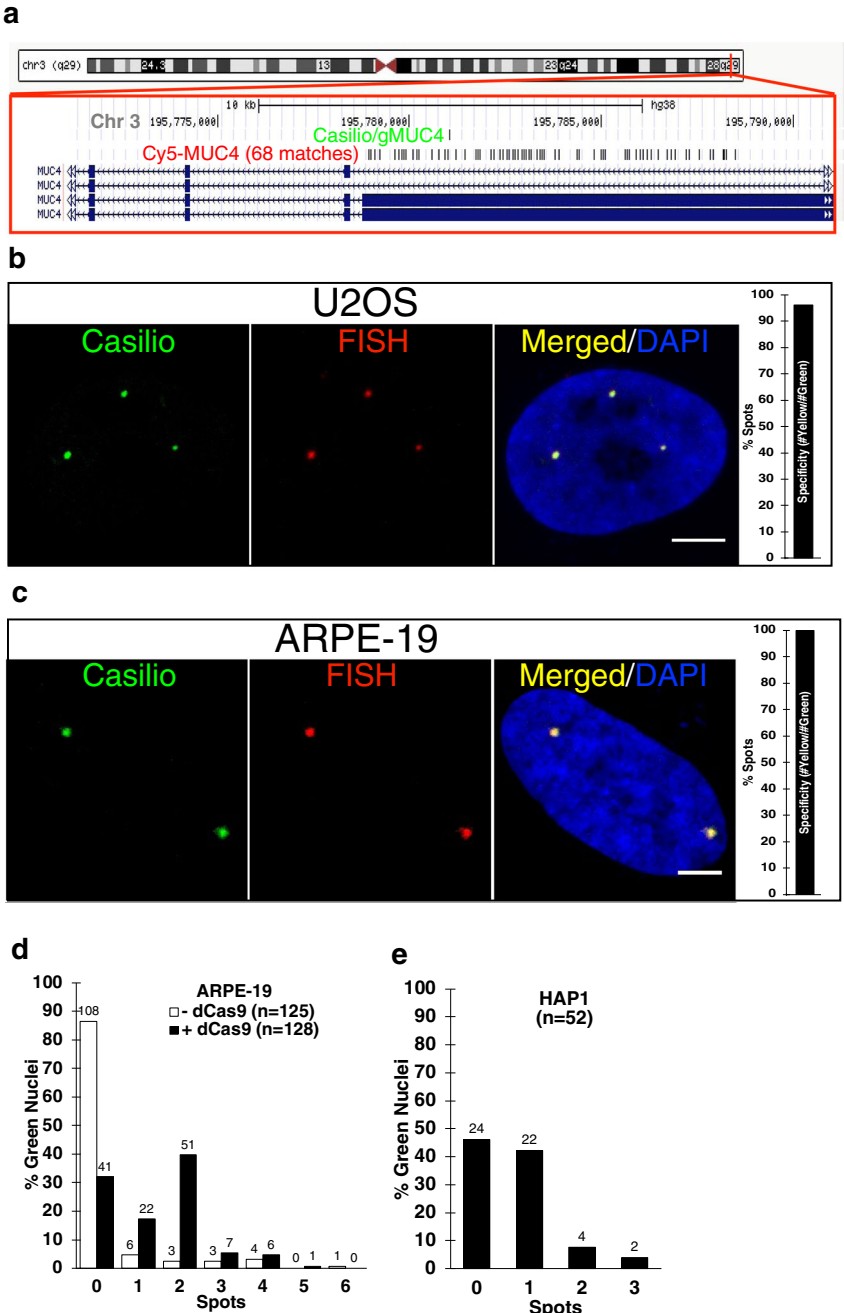

**Fig. 2 Validation of Casilio imaging specificity. a** Schematic showing locations of Casilio *MUC4* gRNA (gMUC4) and Cy5-MUC4 FISH probes. Chr, chromosome. **b** Co-labeling of *MUC4* locus by Casilio (Green) and DNA-FISH (Cy5-MUC4, Red) in U2OS cells ($n = 8$ nuclei; 28 spots; Specificity = #Red-Green-overlap/#Green=96%). Scale bar, 5 μm. **c** Co-labeling of Casilio (Green) and DNA-FISH (Cy5-MUC4, Red) in ARPE-19 cells ($n = 16$ nuclei; 40 spots; Specificity = 100%). Scale bar, 5 μm. **d** Percentage of green ARPE-19 nuclei with the indicated numbers of Casilio/gMUC4 spots in live cells without and with dCas9. Nucleus counts are indicated on top of the columns. **e** Percentage of green HAP1 nuclei with the indicated numbers of Casilio/gMUC4 spots in live cells. Nucleus counts are indicated on top of the columns. Source data are provided as a Source Data file.

(Fig. 4c; Supplementary Video 3; Supplementary Figs. 2c, 5). Depletion of RAD21 led to farther pairwise distances of the Casilio spots in cells treated with auxin compared to the untreated control, with medians of 2.09 μm and 1.21 μm, respectively (Mann–Whitney two-sided $p$-value $= 1.69 \times 10^{-53}$) (Fig. 4d), consistent with the reduced interaction between the promoter and super enhancer of *IER5L* upon RAD21 depletion[29]. We also used Casilio to image a "RAD21-independent" interaction connecting two H3K27ac-enriched super-enhancers and displaying an opposite behavior, in which interaction frequency increased upon RAD21 depletion[29]

(Fig. 4e). Consistent with the observation obtained by sequencing experiments, pairwise distances of Casilio spots decreased upon auxin-mediated RAD21-depletion, with medians of 1.24 μm and 0.57 μm, respectively (Mann–Whitney two-sided $p$-value $= 1.81 \times 10^{-108}$), for the untreated and auxin-treated cells (Fig. 4f–h; Supplementary Videos 4, 5; Supplementary Fig. 2d, e).

These data demonstrate that Casilio can image interaction between cis-regulatory elements to study the contribution of protein factors, such as cohesin, on dynamic chromatin interactions underlying gene regulation.

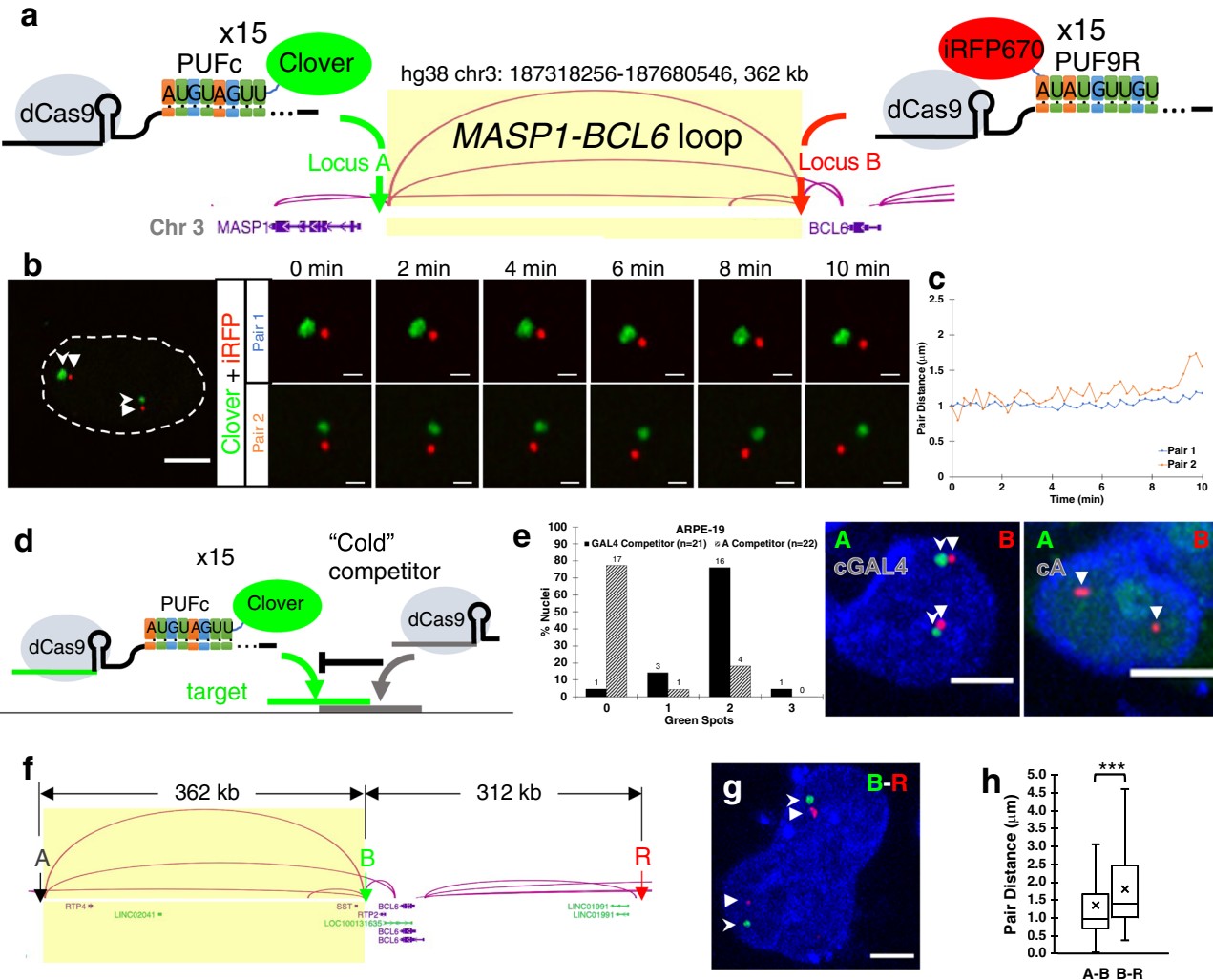

**Fig. 3 Casilio live-cell imaging of chromatin interactions and validation. a** Casilio probes for visualizing chromatin interactions mediated by cohesin. Locus A (5′ anchor) labeled by gRNA-15xPBSc/Clover-PUFc and locus B (3′ anchor) labeled by gRNA-15xPBS9R/PUF9R-iRFP670. *MASP1-BCL6* loop with anchors 367 kb apart. Arc height is proportional to number of ChIA-PET sequencing reads. **b** Representative time-lapse images of *MASP1-BCL6* loop anchors, locus A near *MASP1* (Clover, green, stealth arrowhead) and locus B near *BCL6* (iRFP670, red, triangle arrowhead), in ARPE-19 cells. Image strips show two allelic pairs at indicated time points. Whole nucleus (outlined) at time zero is shown on left. See Supplementary Video 1. Whole nucleus scale bar, 5 μm. Time-lapse scale bars, 1 μm. Experiment repeated 8 times. **c** Pairwise 3D distances of fluorescent foci in **b** over time. See Supplementary Fig. 1 for all distance plots of imaged nuclei (n = 20 nuclei; 33 pairs). **d** Schematic of competitor experiment. "Cold" competitor gRNA without Pumilio binding sites (PBS) for fluorescent protein recruitment binds to sequence overlapping test target. **e** Competitor experiments for validating *MASP1-BCL6*. Column plot shows percentages of nuclei with indicated numbers of Clover (green) spots (nucleus counts on top of columns) in ARPE-19 cells co-transfected with Clover-PUFc/A-15xPBSc, PUF9R-iRFP670/B-15xBPS9R and either a non-competing GAL4 gRNA (black columns) or a gRNA competing with corresponding A locus of the loop (patterned columns). Representative images in non-competed or locus A-competed samples (Clover, green, stealth arrowheads; iRFP670, red, triangle arrowheads). Scale bars, 5 μm. Experiment repeated 3 times. **f** Imaging of downstream region B–R equidistant and adjacent to *MASP1-BCL6* loop (A–B) with locus R at 312 kb 3′ of locus B. Arc height is proportional to number of ChIA-PET sequencing reads. **g** Representative image of co-labeling of locus B (Clover, green, stealth arrowheads) and locus R (iRFP670, red, triangle arrowheads) in ARPE-19 cells (n = 10 nuclei; 14 pairs). Scale bar, 5 μm. Experiment repeated 3 times. **h** Boxplot showing pairwise 3D distances of A–B pair (n = 2347 measurements; mean = 1.36 μm; median = 0.97 μm) and B–R pair (n = 666 measurements; mean = 1.81 μm; median = 1.40 μm) over time. Boxplot displays median line, mean x-marked, interquartile range boxes, and min to max whiskers. ***Mann–Whitney test two-sided p = 6.01 × 10^{−29} < 0.00001. Source data are provided as a Source Data file.

**Visualization of folding dynamics with a three-color palette.** Imaging specific interactions of non-coding elements such as enhancers and promoters provide new insights into gene regulation, while visualizing a continuous stretch of genomic region will improve our understanding of structural folding dynamics and illuminate the process of chromatin loop formation. Given that Casilio can image each nonrepetitive locus with a single gRNA, we next explored the possibility of simultaneously imaging multiple nonrepetitive loci, with the ultimate goal of tracking the

"live" structure of a continuous genomic region. We call this technique, where sequential Casilio probes are deployed across a stretch of genomic DNA, "Programmable Imaging of Structure with Casilio Emitted sequence of Signal" or PISCES.

To increase the number of nonrepetitive loci that can be labelled, we developed a three-color palette. We used three gRNAs (IER5L P-20xPBS_{48107}, M-20xPBSc, and SE-25xPBS9R) (Fig. 5a) for labeling three nonrepetitive locations at the *IER5L* P-SE loop, with a three-color fluorescent protein palette

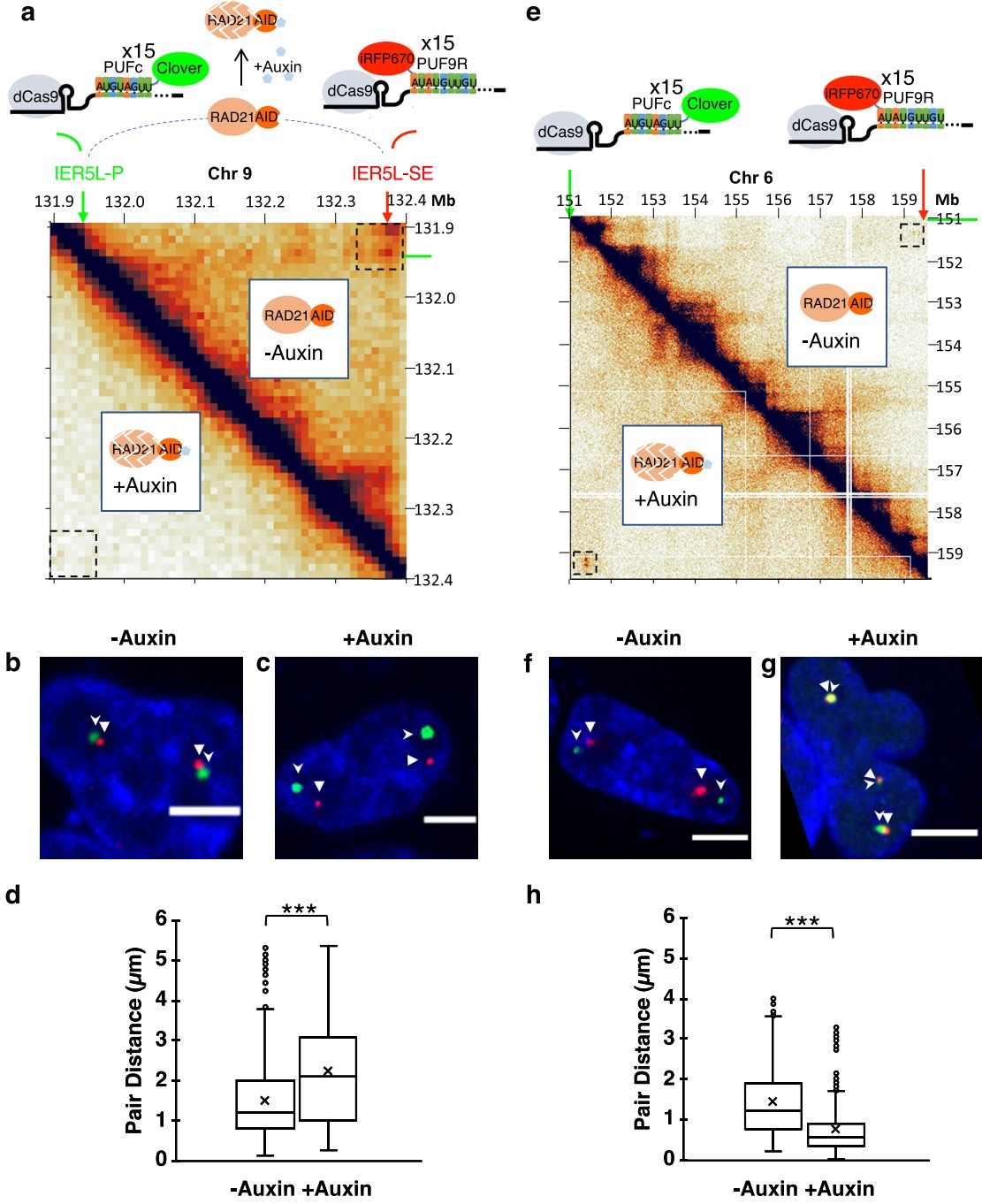

consisting of Clover-PUF$_{48107}$, iRFP670-PUFc, and PUF9R-mRuby2 (Fig. 5b, c; Supplementary Videos 6, 7; Supplementary Figs. 2f, g, 6). These experiments demonstrated the use of dual or triple-color fluorescent protein combinations to encode a sequence of PISCES probes for visualizing multiple reference points along a genomic region.

## Discussion

In this study, we present a CRISPR/Cas-based method for live cell fluorescence imaging of nonrepetitive genomic loci with low gRNA requirement (one gRNA per locus) and high spatiotemporal resolutions. Although gRNA-scaffold methods based on MS2/PP7/boxB hairpins have been attempted before to reduce

gRNA requirement for non-repetitive DNA labeling, the addition of 14 or more of these hairpins to gRNA greatly reduces its expression[18], whereas in Casilio, addition of up to 47 copies of PBS did not affect dCas9 binding activity[23]. The high signal of target-bound Casilio complexes compared to the background unbound complexes can be explained by the observation that gRNA is unstable unless complexed with dCas9 at the target sites[21]. These unique properties of Casilio enable the reduction of its gRNA requirement compared with previously published CRISPR-based approaches and allow imaging experiments to be conducted with standard, widely available microscopy systems. This, in combination with the convenient oligonucleotide-based cloning protocol[30], translates into low-cost genome imaging (in

**Fig. 4 Casilio visualization of RAD21-dependent and -independent interactions in the presence and absence of RAD21. a** Schematic of Casilio imaging of chromatin interactions between *IER5L* promoter (IER5L-P) and a super enhancer (IER5L-SE) in HCT116/RAD21-mAID cells. Hi-C map shows interactions in this region in HCT116/RAD21-mAID cells derived from a previous study[29]. Upper right off-diagonal and lower left off-diagonal plots pairwise interaction between genomic bins in untreated and auxin-treated cells, respectively. Dotted squares mark the area of interest. Chr, chromosome. Mb, megabase. **b**, **c** Representative images of IER5L-P and IER5L-SE labeled by Clover (green, stealth arrowhead), and iRFP670 (red, triangle arrowhead), respectively, in nuclei of HCT116/RAD21-mAID cells without (**b**, see Supplementary Video 2) or with auxin treatment (**c**, see Supplementary Video 3). Scale bars, 5 μm. Experiments repeated 7 times. **d** Boxplot of pairwise spot distances in untreated ($n = 1325$ measurements of 33 pairs from 18 nuclei; mean = 1.52 μm; median = 1.21 μm) and auxin-treated ($n = 1435$ measurements of 35 pairs from 20 nuclei; mean = 2.26 μm; median = 2.09 μm) cells. Boxplot displays a median line, mean x-marked, interquartile range boxes, and min to max whiskers. ***Mann-Whitney test two-sided $p = 1.69 \times 10^{-53} < 0.00001$. See Supplementary Figs. 4, 5 for all distance plots of imaged nuclei. **e** Hi-C map showing the pairwise interaction between genomic bins encompassing a RAD21-independent loop in untreated (upper right off-diagonal) and auxin-treated (lower left off-diagonal) cells. Dotted squares mark the area of interest. **f**, **g** Representative images of the RAD21-independent interaction labeled by Clover (green, stealth arrowhead), and iRFP670 (red, triangle arrowhead), respectively, in nuclei of HCT116/RAD21-mAID cells without (**f**, see Supplementary Video 4) or with auxin treatment (**g**, see Supplementary Video 5). Scale bars, 5 μm. Experiment repeated 4 times. **h** Boxplot of pairwise spot distances in untreated ($n = 1046$ measurements of 27 pairs from 18 nuclei; mean = 1.44 μm; median = 1.24 μm) and auxin-treated ($n = 921$ measurements of 25 pairs from 20 nuclei; mean = 0.76 μm; median = 0.57 μm) cells. Boxplot displays a median line, mean x-marked, interquartile range boxes, and min to max whiskers. ***Mann-Whitney test two-sided $p = 1.81 \times 10^{-108} < 0.00001$. Source data are provided as a Source Data file.

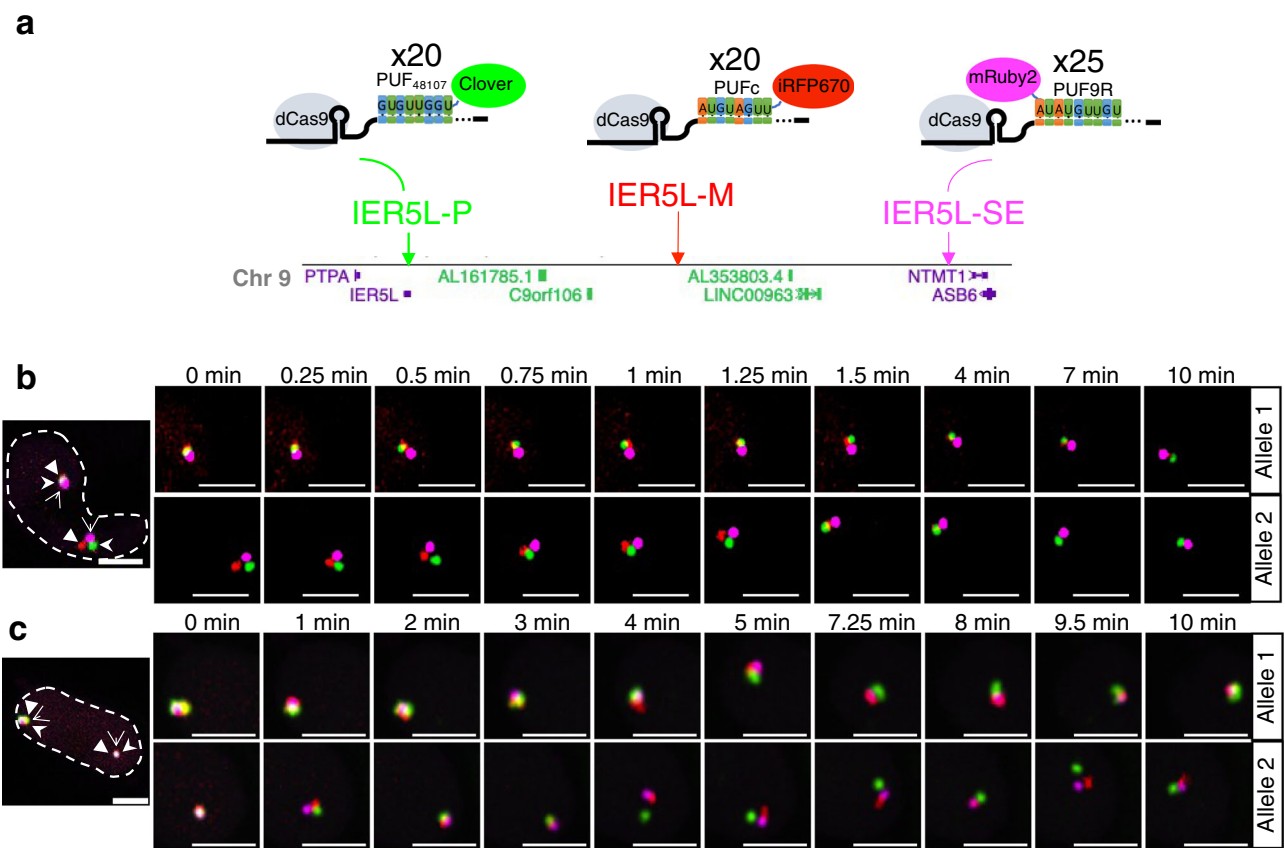

**Fig. 5 Three-color three-point live-cell imaging of chromatin loops. a** Casilio probes for visualizing chromatin interactions of *IER5L* promoter and its super-enhancer, with locus P (promoter) labeled by gRNA-20xPBS48107/Clover-PUF48107, locus M (mid-point) labeled by gRNA-20xPBSc/iRFP670-PUFc, and locus SE (super-enhancer) labeled by gRNA-25xPBS9R/PUF9R-mRuby2. Chr, chromosome. **b**, **c** Representative nuclei (nuclear boundaries outlined) and time-lapse images of *IER5L*-P-SE loop, with *IER5L* promoter, midpoint and super-enhancer labeled by Clover (green, stealth arrowhead), iRFP670 (red, triangle arrowhead), mRuby2 (magenta, open arrowhead), respectively, in HCT116/RAD21-mAID cells at the indicated time point. Scale bars 5 μm. See Supplementary Videos 6, 7. See Supplementary Fig. 6 for distance plots of imaged nuclei ($n = 18$ nuclei; 28 alleles). Experiment repeated 7 times.

our lab, less than USD$5 per target). The "one-gRNA-per-target" requirement not only significantly reduces the technical challenge of using live cell imaging to study chromatin interactions in hard-to-transfect cells, but also it simplifies the design of genome-wide gRNA libraries for imaging. We applied Casilio to visualize the dynamic interactions of DNA elements in native, unmodified chromosomes, in the presence or absence of RAD21, showing RAD21-dependent and independent loops display opposite

behaviors upon RAD21 depletion. By developing a three fluorescent protein palette, we showed that chromatin loops can be imaged at multiple reference points over time. These revealed the highly heterogeneous and dynamic nature of chromatin interactions and structural changes not attainable with bulk cell sequencing approaches, underscoring the need for studying the 4D nucleome with high spatiotemporal resolution in live cells. As proof-of-principle, we visualized the dynamics of a few cis-

regulatory elements in the presence or absence of one of the many proteins involved in the regulation of the 3D genome. Casilio-Imaging should be applied in future studies to track chromatin dynamics over longer time courses upon depletion and replenishment of RAD21, CTCF, WAPL, NIPBL, and PolII, etc.[29,31,32], across a spectrum of cis-regulatory elements (e.g., enhancers, silencers, promoters) in different epigenetic contexts in order to decipher the dynamic molecular processes underlying 3D genome establishment, maintenance and plasticity. One caveat to bear in mind when conducting and interpreting live-cell genome imaging experiments is the potential of the interference of the natural genome folding processes by the binding of the CRISPR reagents, which might be partially alleviated with careful gRNA designs. It is thus an important future goal to systematically evaluate probe design and produce a design pipeline that takes into account the potential interference of the underlying genome folding processes under investigation. Future development of the Casilio imaging toolkit should include higher-density PISCES probe-set with an expanded palette of fluorescent proteins, in order to enable the folding of the genomic region to be delineated at a higher resolution. The multiplexing and expandable nature of Casilio in adapting new effector functions[23], in combination with the advance in genomic imaging demonstrated here, will enable "on-the-fly" and "plug-and-play" perturbation of the (epi)genome (e.g., using Casilio activator, repressor and epigenetic editing modules[23,26]) and concurrent read-out of dynamic 3D chromatin interactions, thereby providing unprecedented flexibility and power for the study of the 4D nucleome (Fig. 1d).

## Methods

**Design of chromatin interaction imaging experiments.** Fastq files of ARPE-19 RAD21 ChIA-PET experiment (ENCSR110JOO) were downloaded from ENCODE database and processed by ChIA-PET2[33]. Interactions were ranked by PET count. Top-ranked interactions with different PET distances were selected. To avoid interference of CTCF binding, design regions were selected by shifting external to PET regions. JACKIE[34] was then used to identify unique 1-copy CRISPR sites in the hg38 genome overlapping design regions of selected loops and further filtered for specificity by Cas-OFFinder[35]. ChIA-PET loops are displayed on WashU EpiGenome Browser [36].

**Cloning for Casilio imaging.** The lentiviral dCas9 expression plasmid (lenti-dCas9-Blast) was generated by PCR-based mutagenesis of lentiCas9-Blast plasmid (Addgene #52962, gift from Feng Zhang). Clover fused with PUF RNA-binding domain was pAC1447 (Clover_PUFc) (Addgene #73689). PUF9R-tethered iRFP670 and mRuby2 were created by a combination of PCR (from IDT gBlocks) and ligation cloning placing iRFP670[37] or mRuby2[38], respectively, downstream of PUF9R. PUFc-tethered iRFP670 and PUF48107 tethered Clover was created by a combination of PCR (from IDT gBlocks) and ligation cloning placing iRFP670[37] upstream of PUFc or Clover upstream of PUF48107, respectively. These PUF-fluorescent protein fusions contain nuclear localization signal (NLS) for their localization in the nucleus.

Guide RNA are under control of the human U6 promoter. gRNA spacer sequences were cloned into sgRNA-PBS expression vectors pCR8-sgRNA-15xPBSc or pCR8-sgRNA-15xPBS9R via an oligonucleotide-annealing protocol [30].

All plasmids were subjected to restriction diagnostic tests and sequenced to ensure correct cloning. Plasmids are deposited on Addgene (Supplementary Table 2).

**Target sequences for gRNAs.** *MUC4*: GCCGGTGACAGGAAGAGTGC
RAD21 loops in ARPE-19 and HCT116/RAD21-mAID cells
*MASP1-BCL6* loop Locus A: GGTAAGAAGCCACTAGGGT
*MASP1-BCL6* loop Locus B: GCATAGCCGCATTTGAAAGC
*MASP1-BCL6* loop Locus R: ACGGATCGGACCCACCATGT
*IER5L*-P: GACTCCCGCGGGTCACTCGG
*IER5L*-SE: GCCATGCTCCGATAAGGATA
*IER5L*-M: GTTTACCCCAAGGGTCGCGG
RAD21-independent loop in HCT116/RAD21-mAID cells
Clover locus: GCTTGCCGACGATGGACCCAT
iRFP670 locus: GAGGATCGAAACGCTGATGCG
Competitor gRNA design
cGAL4: GAACGACTAGTTAGGCGTGTA
c(*MASP1-BCL6*)A (*MASP1-BCL6* loop Locus A competitor, dSaCas9): GTGCTCCAGGGTAAGAAGCCAC

cIER5L-P: GCTTGACTCCCGCGGGTCACT

**Cell culture.** Human osteosarcoma U2OS cells (ATCC® HTB-96™) were cultivated in Dulbecco's Modified Eagle's Medium (DMEM) (Sigma #D5671) with 10% fetal bovine serum (Gibco, ThermoFisher Scientific #10437-028), 4% Glutamax (Gibco, ThermoFisher Scientific #35050-061), 1% Sodium Pyruvate (Gibco, ThermoFisher Scientific #11360-070) and 1% penicillin-streptomycin (Gibco, ThermoFisher Scientific #15140-163). Human retinal pigment epithelial ARPE-19 cells (ATCC® CRL-2302™) were cultivated in DMEM/F12 (Gibco, ThermoFisher Scientific #11330-032) with 10% fetal bovine serum, and 1% penicillin-streptomycin. Human chronic myeloid leukemia HAP1 cells (Horizon™ C859) were cultivated in Iscove's Modified Dulbecco's Medium (IMDM, Gibco, ThermoFisher Scientific #12440-053) with 20% fetal bovine serum and 1% penicillin-streptomycin. The HCT116/RAD21-mAID cell line was a gift from Kanemaki Lab[28]. The cells were cultured in McCoy's 5A medium (ATCC #30-2007) with 10% fetal bovine serum, 2 mM L-glutamine (ATCC #30-2214), penicillin-streptomycin. Incubator conditions were humidified 37 °C and 5% $CO_2$. Cell lines expressing constitutive dCas9 were generated by transducing cells with lentiviruses prepared from a lenti-dCas9-Blast plasmid, followed by Blasticidin selection (Sigma #15205). Cell lines were obtained from trusted sources listed above and were not authenticated in our lab.

**Transfection.** U2OS/dCas9 cells were seeded at density of 100,000 cells/compartment in a 35 mm 4-compartment CELLview™ cell culture dish (Greiner Bio-one, VWR #89125-442) 24 h before transfection. Cells were transfected with 550 ng of sgRNA plasmid containing 15 Pumilio Binding Sites (15xPBS) and 50 ng of Clover-PUF fusion plasmid, using 1.5 µl Lipofectamine 3000 (Invitrogen, ThermoFisher Scientific #L3000075). Media was changed at 24 h post-transfection.

ARPE-19/dCas9 cells were seeded at density of 80,000 cells/compartment in 35 mm 4-compartment CELLview™ cell culture dish 17 h before transfection. Cells were transfected with 400 ng of each sgRNA-15xPBS plasmid, 40 ng of Clover-PUFc fusion plasmid, and 40 ng of PUF9R-iRFP670 fusion plasmid using 1.5 µl Lipofectamine LTX (Invitrogen, ThermoFisher Scientific #15338-100). For competitor experiments, 400 ng of the competitor plasmid was added. Media was changed at 24 h post-transfection.

HAP1/dCas9 cells were seeded at density of 150,000 cells/compartment in 35 mm 4-compartment CELLview™ cell culture dish 30 h before transfection. Cells were transfected with 400 ng of sgRNA-15xPBS plasmid and 40 ng of Clover-PUFc fusion plasmid DNA using 3.5 µl Turbofectin (OriGene #TF81001). Media was changed at 24 h post-transfection.

HCT116/RAD21-mAID/dCas9 cells were seeded at density of 60,000 cells/compartment in 35 mm 4-compartment CELLview™ cell culture dish the day before transfection. For each well, cells were transfected with 300 ng of each sgRNA plasmid and 40 ng of each fluorescent protein plasmid using 3.5 µL TurboFectin. For competitor experiments, 300 ng of the competitor plasmid was added. For three-color experiments, cells were transfected with 400 ng of each sgRNA plasmid, 40 ng of Clover-PUF48107 fusion plasmid, 40 ng of iRFP670-PUFc fusion plasmid, and 400 ng of PUF9R-mRuby2 fusion plasmid using 3.5 µL TurboFectin. Media was changed at 24 h post-transfection. Degradation of the AID-tagged RAD21 was induced by the addition of 500 µM (final concentration) indole-3-acetic acid (IAA/auxin; Sigma Aldrich #I5148) during media change 24 h prior to imaging if applicable.

**Live-cell confocal microscopy.** Imaging was performed at 41–51 h post-transfection. Prior to imaging, cells were stained with 0.5–1.0 µg/ml Hoechst 33342 prepared in cell culture media for 15–30 min, followed by two media washes. Images were acquired with the Dragonfly High Speed Confocal Platform 505 (Andor) using an iXon EMCCD camera and a Leica HCX PL APO ×40/1.10 W CORR objective or a Leica HC PL APO ×63/1.47NA OIL CORR TIRF objective mounted on a Leica DMi8 inverted microscope equipped with a live-cell environmental chamber (Okolab) at humidified 37 °C and 5% $CO_2$. Imaging mode was Confocal 25 µm. Hoechst images were acquired with a 200 mW solid state 405 nm laser and 450/50 nm BP emission filter. Clover images were acquired with a 150 mW solid state 488 nm laser and 525/50 nm BP emission filter. mRuby2 images were acquired with a 150 mW solid state 561 nm laser and 620/60 nm BP emission filter. iRFP670 images were acquired with a 140 mW solid state 637 nm laser and 700/75 nm BP emission filter. Z-series covering the full nucleus was acquired at 0.27 µm step size for ×40 objective or 0.13 µm step size for ×63 objective. For time-lapse imaging, the Z-series was acquired at 0.32 µm step size for ×40 objective or 0.16–0.2 µm step size for ×63 objective. Two-color images were acquired every 15 s. Three-color images were acquired every 15 or 24 s. Images are maximum intensity projection of Z-series. Images were processed in Fusion software using ClearView-GPU deconvolution with the Robust (Iterative) algorithm (pre-sharpening 0, 5 iterations, and de-noising filter size 0.1). Linear adjustments in maximum and minimum levels were applied equally across the entire image to display only nuclear spot signals, and not background nuclear fluorescence. Example images of our image processing pipeline are shown in Supplementary Fig. 2.

**Image analysis.** To measure two targeted genomic loci distance in time-lapse images, Fiji image analysis software was used[39]. Z-series acquired at 0.32 µm step

size for ×40 objective, and 0.16–0.2 μm step size for ×63 objective was used. If the nucleus drifted over time, the Correct 3D Drift plugin was used[40]. For segmenting and tracking spots, the TrackMate plugin was used[41]. Blob diameter was set at 2.0 μm. For each channel, threshold was set to include two spots with the maximum intensity in the 3D volume of the nucleus. Simple LAP Tracker used 2 μm linking max distance, 2 μm gap closing max distance, and 2 gap closing max frame gap. Analysis produced "Spots in track statistics" file which was used to run python script to calculate 3D distances between spots and generate 3D tracks. To determine and correct for potential chromatic aberrations, Invitrogen Tetraspeck 0.1 μm Microspheres (ThermoFisher #T7279) were imaged. On the Dragonfly High Speed Confocal Platform 505 (Andor), Z-series were acquired for both the Leica HCX PL APO ×40/1.10 W CORR objective and the Leica HC PL APO ×63/1.47NA OIL CORR TIRF objective. TrackMate was then used to localize the positions of the microspheres. iRFP670 and mRuby2 chromatic aberrations in each X,Y,Z dimension were calculated relative to Clover, and corrected by offsetting the shifts in each X,Y,Z dimension as follows:

| Objective | Fluorescent channel | X (μm) | Y (μm) | Z (μm) |
|---|---|---|---|---|
| ×40 | iRFP670 | −0.035 | −0.007 | +0.691 |
| ×40 | mRuby2 | −0.053 | −0.055 | +0.319 |
| ×63 | iRFP670 | −0.001 | −0.012 | +0.147 |
| ×63 | mRuby2 | −0.018 | −0.035 | −0.028 |

**DNA-FISH**. Cells were cultured on 12 mm circle coverslip on 24-well plates and transfected with Casilio components. At 48 h, cells were fixed with 4% formaldehyde at room temperature for 10 min, briefly washed once with 1x PBS, permeabilized with 0.5% PBST for 20 min followed by x1 PBS wash, then dehydrated by a series of 75%, 85% and 100% ethanol for 2 min each. Let dry. FISH probe (Cy5-MUC4: /5Cy5/CTTCCTGTCACCGAC, IDT custom DNA oligo) and DNA hybridization buffer were prewarmed and mixed well, added onto slide, covered with cell-attached coverslip, sealed with rubber cement. The sealed cells and probe were then heated on a heat block at 80 °C for 2 min, incubated in a dark humidified chamber at 37 °C overnight. After hybridization, the cell-attached coverslip was unsealed and washed in sequence with ×2 SSC at room temperature for 2 min, 10% formamide/×2 SSC for 5 min at 45 °C, ×2 SSC at room temperature for 5 min, ×0.2 SSC at 45 °C for 10 min twice, then counterstained with DAPI for 5 min and mounted with SlowFade™ Diamond Antifade Mountant (Invitrogen, Thermofisher Scientific #S36967) for imaging.

**Genome track and Hi-C maps**. Genome tracks were visualized by WashU Epi-Genome Browser[36] or UCSC Genome Browser[42]. Hi-C maps were generated using FAN-C[43].

**Statistics and reproducibility**. No statistical method was used to predetermine sample size. Sample sizes were chosen as customary in the field and were sufficient to obtain statistical significance. No data were excluded from the analyses. The experiments were not randomized. The Investigators were not blinded to allocation during experiments and outcome assessment.

**Reporting summary**. Further information on research design is available in the Nature Research Reporting Summary linked to this article.

## Data availability

All data are available in the main text, the Supplementary Materials and data repository. Plasmids are available on Addgene (see Supplementary Table 2 for plasmid listing with links). Raw images[44] are deposited on FigShare at https://doi.org/10.6084/m9.figshare.19317977. Previously released ARPE-19 ChIA-PET data are available at ENCODE database with accession number ENCSR110JOO. Previously published HCT116/RAD21-mAID HiC datasets[29] are available at GEO database with accession number GSE104334. Source data are provided with this paper.

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

## Acknowledgements
We thank the Jackson Laboratory Microscopy Service managed by Qianru Yu for microscope usage, Erick Ratamero (Research IT Applications) for imaging analysis support, Jane Cha (JAX Creative) for graphic design, and Research Program Development for help with manuscript editing. This work is partially supported by internal funds provided by the Jackson Laboratory, Arizona State University as well as grants from the National Human Genome Research Institute (R01-HG009900 to A.W.C.), the National Science Foundation (CCF-1955712 to A.W.C), and the National Cancer Institute (P30CA034196 to A.W.C.).

## Author contributions
J.J.Z. and A.W.C. conceived the study. P.A.C., J.J.Z and A.W.C. designed the experiments. P.A.C., M.D., N.J., A.T., J.J.Z., and A.W.C. performed the experiments. P.A.C and M.D. analyzed the data. P.A.C., A.T., J.J.Z., and A.W.C. wrote the manuscript.

## Competing interests
Patent application by the Jackson Laboratory with no. PCT/US2020/046076 with P.A.C and A.W.C on the inventorship is pending on the Casilio imaging method. The remaining Authors declare no competing interests.
