## [Peer Review File · Nature Communications]

Reviewers' Comments:

Reviewer #1:

Remarks to the Author:

Previously, Dr. Cheng's group further developed the Casilio system, in which they took advantage of the PUF domains for efficient targeting and delivering. In this work, the major update is that now the system can target non-repetitive region, which is a big advancement. Such strategy can make the live cell imaging system works for a lot more distal elements, such as TF binding sites or enhancers. They also used multiple examples to demonstrate the feasibility of this system and the results are convincing. The three-color system in Fig. 5 is also very interesting. Below are my suggestions/comments:

1. Most importantly, could the authors clarify where are the other systems out there for comparisons? A table would be really helpful for the readers. Note I don't think a method/software needs to beat all other tools in all aspects to be published. I believe Casilio is a unique system, but we need to give a fair assessment of all other systems if they exist;
2. Fig. 3 is very nice. In FISH, we usually use the region on the other size of the gene promoter (equal distance) as control. Is that doable?
3. Fig. 4, RAD21 independent loops are very interesting. What are they? Any other histone modifications or TF binding? I understand this loci was reported in another paper, but it would be great for the authors to give a much more detailed description of this loop. Adding some hypothesis of the mechanism would also be stimulating.
4. Can the authors discuss the realistic throughput/cost of the Casilio system. In particular, I am interested to learn how challenging it would be for a regular molecular biology lab to adopt this system.

Reviewer #2:

Remarks to the Author:

This study by Clow and colleagues presents a repurposing of the Casilio platform for imaging. Importantly the authors convincingly show that a single gRNA recruits enough fluorescent molecules to allow live imaging of single loci. This is an important technological advancement that could provide important advances to the field.

I have the following suggestions for improvement of the study.

- 1- The title might be more appealing if it highlights that it allows using a single gRNA for imaging. For example: CRISPR-mediated Multiplexed Live Cell Imaging of Nonrepetitive Loci with a single guide RNA.
- 2- I find that figure 1 could be improved. Right now, figure 1 A does not provide a good representation of how Pumilio works. For example, what is X, Y A and Z? How many repeats will actually be used in this study? Why not actually show already in figure 1 the schemes that will later be used in figure 5? In contrast, figure 1B occupies much more space while not explaining the actual attributes of the system shown in this study. It is a figure more suitable for a review.
- 3- It is a little confusing why the Muc4 gene, known for being highly repetitive is used to illustrate the advantage of this system to label single loci with one gRNA. Is this because then it can be compared to FISH using a single oligo? If that is the reason it should be clearly stated. Otherwise it may be confusing why a repetitive locus was used.
- 4- Figure 3, locus A (clover) shows two alleles with very different signal size. Any potential explanation? It would be important to comment on this.
- 5- Since the gene names cannot be read in figure 4, wouldn't it be better to remove the side annotations surrounding the HiC matrices and just place the location of the promoter and super enhancer being investigated, together with the genomic coordinates.
- 6- It would be a really impressive addition to the study if the authors could track the changes in distance shown in figure 4 as cohesin is degraded and then following auxin washoff. Currently there are only estimates of how long loops take to be established. This would actually allow one to see them. It would be a very good demonstration of the strength of this system for live imaging. It

would also show for how long live imaging can be done with this system as all figures show labeling just for a few minutes. Can loci be tracked through longer periods of time?

Response to Reviewers' comments

We would like to thank the reviewers for their constructive feedback to improve our manuscript. Below we provide point-by-point responses to reviewers' comments and highlight changes made to the manuscript following reviewers' suggestions.

Reviewer #1 (Remarks to the Author):

Previously, Dr. Cheng's group further developed the Casilio system, in which they took advantage of the PUF domains for efficient targeting and delivering. In this work, the major update is that now the system can target non-repetitive region, which is a big advancement. Such strategy can make the live cell imaging system works for a lot more distal elements, such as TF binding sites or enhancers. They also used multiple examples to demonstrate the feasibility of this system and the results are convincing. The three-color system in Fig. 5 is also very interesting. Below are my suggestions/comments:

1. Most importantly, could the authors clarify where are the other systems out there for comparisons? A table would be really helpful for the readers. Note I don't think a method/software needs to beat all other tools in all aspects to be published. I believe Casilio is a unique system, but we need to give a fair assessment of all other systems if they exist;

Thank you for the reviewers' suggestion. It would be a great addition to the manuscript and help readers immediately understand the differences between systems. As suggested, a table summarizing and comparing previously published systems with Casilio is added as **Supplementary Table 1**.

2. Fig. 3 is very nice. In FISH, we usually use the region on the other size of the gene promoter (equal distance) as control. Is that doable?

We have new imaging data with an external probe "R" equidistant and downstream of the imaged *MASP-BCL6* loop anchor B (**Fig 3f-h**). Consistent with ChIA-PET data which showed lower degree of interaction between B-R than the *MASP-BCL6* loop (A-B), we observed longer physical 3D distances between B-R than A-B in imaging.

3. Fig. 4, RAD21 independent loops are very interesting. What are they? Any other histone modifications or TF binding? I understand this loci was reported in another paper, but it would be great for the authors to give a much more detailed description of this loop. Adding some hypothesis of the mechanism would also be stimulating.

We agree that the RAD21-independent loops are very interesting. The Rao et al 2017 *Cell* paper describes those as super enhancers highly enriched with H3K27ac and DNA binding proteins. The RAD21-independent interactions also occur interchromosomally. The depletion of RAD21 increases interactions of these elements without changes in H3K27ac level (Rao, 2017). We are guessing that in the absence of RAD21, these elements may colocalize through some

kind of phase separation/condensation processes due to the high H3K27ac level, perhaps in a non-specific manner. In the presence of RAD21, the “normal” looping processes mediated in part by RAD21 organize these super-enhancers specifically with their target genes within transcription hubs and thus preclude them from interacting with each other. However these are our speculation only without strong supporting evidence. Unfortunately, our lab is not expert in phase separation or epigenetics and we currently lack further data to generate an otherwise sound, non-speculative hypothesis underlying the mechanisms of these RAD21-independent interactions. It would be exciting in the future to collaborate with experts in phase separation, epigenetics and 3D genome to decipher the mechanisms using a combination of conditional factor knockout/depletion (WAPL, NIPBL, RAD21, PolII, CTCF), Casilio-mediated epigenetic editing (e.g., removing the H3K27ac mark specifically at these elements) and concurrent Casilio-mediated genome imaging, super-resolution single-molecule transcription factor imaging, omics/sequencing methods, as well as biophysical approaches for investigating phase separation/condensates.

4. Can the authors discuss the realistic throughput/cost of the Casilio system. In particular, I am interested to learn how challenging it would be for a regular molecular biology lab to adopt this system.

We have ongoing collaborations with other labs to image more loops. From our experience, if we design 3 gRNAs per anchor, there will be at least one pair that works efficiently. The cost is low because the cloning is easy through an oligo annealing protocol, and it only needs one gRNA per target. Each gRNA plasmid costs us less than USD\$5 to clone. Even if we need to test three gRNAs per anchor, for a pair of contacts, we only will need to clone a total of 6 plasmids (~\$30). Transfection is also a standard method. The live cell confocal microscopy system we use is also a quite ubiquitous commercial model. We added some description about cost and throughput in the last paragraph/discussion section.

Reviewer #2 (Remarks to the Author):

This study by Clow and colleagues presents a repurposing of the Casilio platform for imaging. Importantly the authors convincingly show that a single gRNA recruits enough fluorescent molecules to allow live imaging of single loci. This is an important technological advancement that could provide important advances to the field.

I have the following suggestions for improvement of the study.

1- The title might be more appealing if it highlights that it allows using a single gRNA for imaging. For example: CRISPR-mediated Multiplexed Live Cell Imaging of Nonrepetitive Loci with a single guide RNA.

Thank you very much for the reviewer’s suggestion. We have changed the title to “CRISPR-mediated multiplexed live cell imaging of nonrepetitive loci with one guide RNA per locus”

2- I find that figure 1 could be improved. Right now, figure 1 A does not provide a good representation of how Pumilio works. For example, what is X, Y A and Z? How many repeats will actually be used in this study? Why not actually show already in figure 1 the schemes that will later be used in figure 5? In contrast, figure 1B occupies much more space while not explaining the actual attributes of the system shown in this study. It is a figure more suitable for a review.

We have updated figure 1 to hopefully make the introduction of Casilio better. We would like to retain figure 1B (now Figure 1d) to present an exciting potential usage scenario combining Casilio-Imaging presented in this paper as well as our previously published epigenetic editing modules to edit epigenetic marks at defined loci and concurrently image changes in chromatin interaction dynamics.

3- It is a little confusing why the *Muc4* gene, known for being highly repetitive is used to illustrate the advantage of this system to label single loci with one gRNA. Is this because then it can be compared to FISH using a single oligo? If that is the reason it should be clearly stated. Otherwise it may be confusing why a repetitive locus was used.

MUC4 gene was used to start off the study because it was used in the first CRISPR-based imaging paper (Chen et al, 2013). It harbors non-repetitive sequences as well as repetitive sequences that allow convenient DNA-FISH with a single oligonucleotide probe binding to a repetitive sequence occurring 68 times in the *MUC4* gene) for co-labeling validation. We have updated the main text to make clear the reasoning behind our selection.

4- Figure 3, locus A (clover) shows two alleles with very different signal size. Any potential explanation? It would be important to comment on this.

Size of foci are different potentially due to the different accessibility of the alleles or due to the stochasticity of the fluorescent protein recruitment. Different sizes and shapes of foci have been seen in other CRISPR imaging publications (e.g., [10.1038/ncomms14725](https://doi.org/10.1038/ncomms14725); [10.1093/nar/gkz752](https://doi.org/10.1093/nar/gkz752); [10.1126/science.aax7852](https://doi.org/10.1126/science.aax7852); [10.1038/s41594-017-0015-3](https://doi.org/10.1038/s41594-017-0015-3); [10.1126/science.aao3136](https://doi.org/10.1126/science.aao3136)) and thus may represent a general phenomenon of live-cell genome imaging or CRISPR-based imaging.

5- Since the gene names cannot be read in figure 4, wouldn't it be better to remove the side annotations surrounding the HiC matrices and just place the location of the promoter and super enhancer being investigated, together with the genomic coordinates.

We greatly appreciate the reviewer's suggestion to make the figure cleaner and have improved the figure as suggested.

6- It would be a really impressive addition to the study if the authors could track the changes in distance shown in figure 4 as cohesin is degraded and then following auxin washoff. Currently there are only estimates of how long loops take to be established. This would actually allow one

to see them. It would be a very good demonstration of the strength of this system for live imaging. It would also show for how long live imaging can be done with this system as all figures show labeling just for a few minutes. Can loci be tracked through longer periods of time?

We would like to thank the reviewer for suggesting this very interesting experiment. However, it is practically challenging for us to complete this experiment in a reasonable amount of time due to my (Albert Cheng's) recent move to a new institution and the lags in lab renovation, equipment purchase, compounded with backordered supply chain caused by the pandemic. The first and other authors moved to other labs or graduated. We would like to propose that we mention these exciting future experiments in the manuscript (**last paragraph/discussion section**). We think the field will truly benefit from a broad systematic study of chromatin interaction dynamics using Casilio to image hundreds/thousands of loops over the RAD21 depletion/reestablishment time course. We will look at the same for other looping factors such as WAPL, NIPBL, CTCF, in different cell types as well as normal/pathological states. And it is our hope that the publication of this method paper would allow us and others to start systematically investigating loop dynamics through live-cell imaging in many different scenarios to obtain exciting new biological insights, and we are really excited for it to soon reach a broad audience in the field!

Reviewers' Comments:

Reviewer #1:

Remarks to the Author:

This is a very nice paper and the authors have addressed all my previous concerns. I would recommend the publication of this manuscript at its current shape.

Reviewer #2:

Remarks to the Author:

The authors have addressed my concerns and in my opinion the article is ready for publication at Nature Communications.